# EEG-Based Emotion Recognition Using Convolutional Recurrent Neural Network with Multi-Head Self-Attention

Zhangfang Hu [1], Libujie Chen [1,2,*], Yuan Luo [1,2] and Jingfan Zhou [1]

[1] Key Laboratory of Optoelectronic Information Sensing and Technology, Chongqing University of Posts and Telecommunications, Chongqing 400065, China
[2] School of Advanced Manufacturing Engineering, Chongqing University of Posts and Telecommunications, Chongqing 400065, China
* Correspondence: s200431009@stu.cqupt.edu.cn

**Featured Application: The proposed method in this study can be used in EEG emotion recognition and achieve better results.**

**Abstract:** In recent years, deep learning has been widely used in emotion recognition, but the models and algorithms in practical applications still have much room for improvement. With the development of graph convolutional neural networks, new ideas for emotional recognition based on EEG have arisen. In this paper, we propose a novel deep learning model-based emotion recognition method. First, the EEG signal is spatially filtered by using the common spatial pattern (CSP), and the filtered signal is converted into a time–frequency map by continuous wavelet transform (CWT). This is used as the input data of the network; then the feature extraction and classification are performed by the deep learning model. We called this model CNN-BiLSTM-MHSA, which consists of a convolutional neural network (CNN), bi-directional long and short-term memory network (BiLSTM), and multi-head self-attention (MHSA). This network is capable of learning the time series and spatial information of EEG emotion signals in depth, smoothing EEG signals and extracting deep features with CNN, learning emotion information of future and past time series with BiLSTM, and improving recognition accuracy with MHSA by reassigning weights to emotion features. Finally, we conducted experiments on the DEAP dataset for sentiment classification, and the experimental results showed that the method has better results than the existing classification. The accuracy of high and low valence, arousal, dominance, and liking state recognition is 98.10%, and the accuracy of four classifications of high and low valence-arousal recognition is 89.33%.

**Keywords:** EEG; emotion recognition; CNN; BiLSTM; multi-head self-attention; time–frequency map



## 1. Introduction

Emotion plays an important role in daily human life and influences all aspects. Emotion is an indispensable and important role for humans that affects people all the time, including human decision-making, speech, sleep, health, communication, and various other characteristics. Emotion recognition is often implemented based on facial expressions, speech, and physiological signals, while physiological signals more accurately reflect fluctuations in human emotional states [1] and are often used in the field of human-computer interaction. In recent years, emotion recognition based on electroencephalography (EEG) [2] in physiological signals has had widespread applications because of its non-invasive, easy-to-use, and inexpensive characteristics.

EEG is an electrical signal of the human brain epidermis, which has nonlinear and non-smooth characteristics, while feature extraction and classification of such signals have been a challenge for researchers. Many researchers have proposed their feature extraction algorithms based on traditional methods [3–6] emotion recognition methods, and deep learning emotion recognition methods based on convolutional neural networks [7], deep

learning networks [6,8], long and short-term memory networks [9–12], graph convolutional networks [9,12,13], attention mechanisms [14], etc. In recent years, graph convolutional neural networks (GCNN), long short-term memory neural networks (LSTM), and attention mechanisms have begun to be applied increasingly in the field. However, we are facing the problem of how to improve the integration of these different functional networks and apply them to EEG-based emotion recognition.

To address the above problems, we propose a deep learning model-based emotion recognition method. The method first uses common spatial pattern to process the EEG signal to get better spatial information, and then continuous wavelet transform transforms the filtered signal into a time–frequency map as the input of the deep model, which incorporates convolutional aspects. The model integrates convolutional neural networks and bidirectional long- and short-term memory networks and adds a multi-headed self-attentive mechanism on top of that. In the hybrid model, feature information is extracted using multiple CNNs, future, and past time series information is learned, and temporal features are extracted using BiLSTM, while better feature information is given by assigning weights to sentiment feature information using a multi-headed self-attentive mechanism. Finally, we conducted extensive experiments on DEAP dataset and the experimental results showed that the method has better classification performance than the existing classification methods.

The main contributions of this paper are as follows:

We propose a deep learning emotion recognition method for EEG signals, which is a deep learning framework consisting of a hybrid model of CNN, BiLSTM, and MHSA. The CNN is used to smooth and down-sample the data, then the BiLSTM extracts the future and past emotion features from the data for learning, and finally, the multi-headed attention mechanism is implemented to improve the emotion recognition accuracy by weighing the key information of the EEG emotion features.

- We propose a method for converting EEG signals into time–frequency maps as model data input. The method uses CSP filtering to extract the spatial information of EEG signals, and then CWT is used to convert the filtered information into time–frequency maps as model input data, which makes the CNN-BiLSTM-MHSA better for extracting feature information.
- We conducted extensive experiments on the DEAP dataset and achieved higher accuracy contrasted to other deep learning models and traditional methods for both binary classification tasks of valence, arousal, dominance, and liking, as well as quadruple classification tasks of valence-arousal states.

The content of the remainder of this paper is arranged as follows: The first part introduces some related studies on EEG sentiment recognition, and the second part describes the proposed sentiment recognition method and its key components, including CNN, BiLSTM, and multi-headed self-attentive mechanism fusion models and their architectures and evaluation metrics, the third part presents the DEAP dataset used in the experiments and analyzes the experimental results of CSP-CWT time–frequency maps and CNN-BiLSTM-MHSA on DEAP in detail, and the fourth part is the summary.

## 2. Related Work

Deep learning is a key element in the research of EEG emotion recognition, and more and more excellent models are being applied to EEG emotion recognition with more in depth research in the field of deep learning. The ability of convolutional neural networks to learn local smooth structures and evolve them into multi-scale hierarchical patterns has led to great breakthroughs in processing tasks such as images, speech, and video [15]. Tabar [8] used a one-dimensional CNN to process temporal, frequency, and location information for EEG extraction. Ozdemir [9] converted EEG signals into multi-spectral topological image sequences and used a two-dimensional convolutional network for feature extraction. Tripathi [16] undertook experiments on the DEAP dataset using CNN with good results. Li et al. [17] used a hierarchical convolutional neural network (HCNN)

to verify the feasibility of this approach on the SEED dataset compared with traditional methods. Cui et al. [12] deeply integrated the complexity of the EEG signal, the spatial structure of the brain, and the temporal context of emotion formation to obtain a 4D feature tensor fed into CNN to extract features and obtained better results. Li et al. [18] used CNN to handle a four-category emotion task. Convolutional neural networks have a great role in the feature extraction of EEG signals. One-dimensional convolutional networks are suitable for the direct processing of EEG signals [8,17], but in the processing of information about images [9,12,16,18], two-dimensional convolution is more effective.

Recurrent neural networks (RNN) with good addressing capability for time series data are commonly used in natural language processing. In the development of deep learning, LSTM, which can better handle long-term dependencies, has become an effective scalable model for recurrent neural networks to solve the learning problem of sequence data. Ozdemir et al. [9] used LSTM to process the features extracted from CNN and achieved better results on the DEAP dataset. The accuracy was 90.62% in the high and low valence test, 86.13% in the high and low arousal test, and 86.23% in the liking test. Li et al. [18] used LSTM to fuse the spatial, frequency domain, and temporal features of the original EEG signal to make progress in a four-classification task. Many researchers currently are starting to use BiLSTM networks more and more, due to their ability to learn past and future information in time series and get better results than LSTM. Fares et al. [10] proposed a bidirectional neural network for EEG-based image recognition. They used bi-directional long short-term memory (BiLSTM) as a feature encoder and then used independent component analysis (ICA) and support vector machine (SVM) to classify the features which achieved advanced results. Xie et al. [11] proposed an end-to-end BiLSTM model with a neural attention mechanism for EEG-based target recognition tasks. Cui et al. [12] used BiLSTM to serialize feature-specific information processed by CNN and then learn the relationship between past and future in the information, achieving significant improvement with an average accuracy of 94% in the DEAP dataset. Sharma et al. [19] used the BiLSTM model for experimental validation on both dichotomous and quadruple classification, with significant room for improvement on the quadruple classification task. In terms of model selection for dealing with EEG emotion recognition. The use of Conv-LSTM in the literature [20] has a good performance in processing time series, and advanced results were obtained on all three datasets. BiLSTM has more obvious advantages over LSTM for learning emotional information. The model proposed in this paper also utilizes the features of BiLSTM to achieve good results in EEG emotion recognition.

Attention mechanisms are often used to process natural language [21], where the relationship between the input and the word to be predicted is attended to in the sentence where the encoder output is obtained, and attention weights are calculated to determine the part of the input data that has the greatest impact on the output data, and then this part is given a higher weight, while this part of the input can have a greater value in the network. With the use of deep learning in the field of EEG emotion recognition, the attention mechanism has improved the ability to recognize emotional information. Kim et al. [14] used the attention mechanism to assign weights based on peaks to the emotional states that occur at a particular moment in a three-level emotion classification study up to 91.8%. Nowadays, self-attention mechanisms are less frequently used in EEG emotion recognition to improve classification accuracy, while multi-headed self-attention mechanisms have a greater advantage in this task of emotion recognition. In this study, the addition of multi-headed self-attention mechanism to the fusion model used in this paper significantly improved the accuracy of emotion recognition.

The mixture of traditional methods and machine learning in EEG emotion recognition also has good results. Pandey et al. [6] used wavelet transform as a feature extraction method to form EEG signals of different frequency bands and used DNN for binary classification of emotional states; however, the results were not good. In the literature [22–24], wavelet transform was used to decompose time–frequency features and smoothen the feature information into SVM for classification, which shows much improvement in the

accuracy of EEG emotional state recognition in the field of machine learning. Xu et al. [25] used discrete wavelet transform to process the EEG signal and then used the model of mRMR-KELM to achieve 80.83% accuracy in quadruple classification emotion classification recognition. Sharma et al. [19] used discrete wavelet transform to decompose the EEG signal into sub-bands and particle swarm optimization to remove irrelevant information. Galvão et al. [13] used wavelet transform to extract EEG features with KNN, and RF was determined as the best machine learning method for regression to achieve 84.4% accuracy in four classifications. The above-mentioned wavelet-transformed features are more conducive to the understanding of the model. In this paper, we also use the continuous wavelet transform to obtain the time–frequency features as the model input, which has better results compared with the direct input of the EEG signal.

Although the above recognition methods have made some progress in EEG emotion recognition, the accuracy of many classifications is relatively low. The successful applications of convolutional neural networks, bidirectional long- and short-term memory networks, and attention mechanisms have provided new ideas for EEG-based emotion recognition. Therefore, we tried a new deep-learning model of emotion recognition to improve its accuracy of emotion recognition.

## 3. Methods

### 3.1. Common Spatial Pattern Filtering

When acquiring EEG signals from the scalp, different channel acquisitions can make the spatial resolution of the signal poor. Therefore, spatial filters are needed to increase spatial resolution and improve the spatial information contained in the signal [26]. In this paper, the CSP algorithm is used as a spatial filter to convert the original time series into a new time series with variance discriminative information and maximize the variance according to the labels to achieve spatial discrimination of two and four class time series.

The spatially filtered signal $Z_{tr}$ for trial tr with dimension N × ch, where N is the number of samples and ch is the number of channels, is determined by:

$$Z_{tr} = W X_{tr}, \tag{1}$$

where $X_{tr}$ is the original EEG signal and $W$ is the projection matrix. The columns of the spatial filter $W^{-1}$ are the spatial patterns, which are the EEG source distribution vectors. The first and last columns are the significant spatial patterns that explain the maximum variance of one class of sentiment and the minimum variance of the other class. The EEG features with the greatest correlation between the two classes of emotions in the EEG are extracted by common-space pattern transformation, in preparation for the next step of the continuous wavelet transformation.

### 3.2. Generation of Continuous Wavelet Transform

The continuous wavelet transform is used to generate a function scale map of frequency and time for better frequency localization of low-frequency and long-term events. In the context of image classification, the images obtained from the scale map are used as input to the network model [27]. In this paper, CWT is used because it produces good time–frequency analysis and helps in the localization of frequency information, which is determined by:

$$W_s(a, \tau) = \frac{1}{\sqrt{a}} \int s(t) \phi^*(\frac{t - \tau}{a}) dt, \tag{2}$$

where $s(t)$ is the input signal, $a$ is the wavelet transform scale, $\phi$ is the wavelet basis function, and $\tau$ is the time shift. The wavelets were compressed according to the resolution by translating along time $t$, with s as the scale. In the spectrogram analysis of EEG signals, the classification of emotional states with a fixed window size is generally effective. In contrast, the multi-resolution analysis based on the scale map can capture short and abrupt changes in EEG signal frequency.

### 3.3. Data Representation Time–Frequency Map

We used the "pywt" library of Python with 14 wavelet families, according to the experimental comparison of "cmor4-4", "gaus8", "cgau8", "mexh", "morl" wavelet families generated by the time–frequency map. "cgau8", "mexh" and "morl" wavelet families. According to the effect of feeding into CNN, we applied the continuous wavelet transform of the "cgau8" wavelet family. The scalar map was plotted as a function of frequency and time to allow better temporal localization of short and high-frequency events and better frequency localization of low frequencies. The frequencies were sampled from 0 to 64 (128 Hz/2 = 64 Hz) for all possible frequencies, and half of the sampling rates we analyzed are given according to Nyquist theory. By segmenting the filtered signal in 6 s, it can be divided into 10 segments, and the CNN input image size of 32 channels of EEG data is $20 \times 65 \times 3$, which is stored according to different label categories. This is the size we confirmed by repeated experiments, too high or too low will make the CNN processing of features worse, while the generated partial images are shown in Figure 1.



**Figure 1.** Time–frequency diagram of CSP filtered signal.

### 3.4. A CNN-BiLSTM Method with Multi-Head Self-Attention

The training model studied in this paper is a bidirectional two-layer LSTM with a multi-headed self-attentive mechanism coupled with a single-layer convolutional neural network structure, referred to as a CNN-BiLSTM-MHSA network. The input data is passed through a one-dimensional convolutional network with a convolutional kernel of 128. Then after batch normalization (BN) and ReLu activation function and then one-dimensional pooling of the data with the kernel of 3, the obtained data sequence is fed to the BiLSTM deep network with 256 hidden units per layer of the LSTM network. The output structure is spliced with fully connected layers, and the 512 data sequences obtained are cloned to the multi-headed self-attentive Q, K, and V, respectively, and then weights are assigned by scaling dot product attention. Finally, the classification results are derived using the Softmax activation function, while the network classification framework diagram is shown in Figure 2.

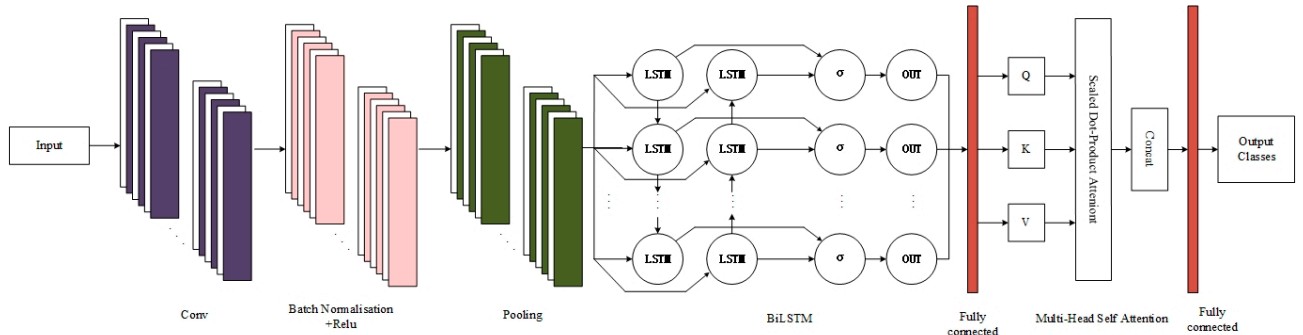

**Figure 2.** The proposed CNN-BiLSTM-MHSA classification framework.

#### 3.4.1. Convolutional Neural Network

CNN is a very effective image processing and classification model. the architecture uses convolutional operations to extract various features of the data and then pass the features to the next layer. The convolutional smoothed signal after CNN processing has a significant improvement on the LSTM network.

### 3.4.2. Bidirectional LSTM Network

LSTM occupies a place in the field of sequential signal analysis by sharing weights, and the weights between its hidden layer and output layer can be recycled at any time [28]. It is a chain model for processing time series and can effectively compensate for the vanishing gradient problem. Compared with the classical unidirectional EEG signal extraction method, the bidirectional EEG signal extraction method can extract dynamic information from both earlier and later segments of the EEG signal sequence [29]. An LSTM unit consists of three gate control units of the forget gate and input gate and the calculation formulas are defined by Equations (3)–(8):

$$f_t = \sigma(W_f \cdot [h_{t-1}, x_t]) + b_f), \tag{3}$$

$$i_t = \sigma(W_i \cdot [h_{t-1}, x_t]) + b_i), \tag{4}$$

$$\widetilde{C}_t = \tanh(W_C \cdot [h_{t-1}, x_t]) + b_C), \tag{5}$$

$$C_t = f_t \times C_{t-1} + i_t \times \widetilde{C}_t, \tag{6}$$

$$O_t = \sigma(W_O \cdot [h_{t-1}, x_t]) + b_O), \tag{7}$$

$$h_t = \tanh(C_t) \times O_t, \tag{8}$$

where, $x_t$ is the time series at time $t$, $C_t$ is the cell state, $\widetilde{C}_t$ is the temporary cell state, $\sigma$ is the sigmoid function, $W$ is the weight matrix, $b$ is the bias vector of the corresponding weight, $h_t$ is the hidden state, $f_t$ is the forget gate, $i_t$ is the memory gate, and $O_t$ is the output gate. The forget gate selects the retained features and inputs both the information of the previous state and the current state information into the Sigmoid function. The memory gate is responsible for updating the state of the LSTM cell, after which the input gate controls the output value to the next LSTM cell. Unlike the above single LSTM, the output formulas of the bidirectional LSTM are defined by Equation (9):

$$y_t = \sigma(W_h \cdot [h_t, h'_t]) + b_h), \tag{9}$$

The BiLSTM network adds a backward layer to learn the future sentiment information, which is an extension of the past sentiment information. The BiLSTM merges the gating architecture and bidirectional characteristics perfectly so that more information can be remembered and processed by the two LSTM units [30]. The network structure of BiLSTM is shown in Figure 3. The time series is input to the model, the forward layer connects the feature information in the past series with the present information, the backward layer connects the future, and finally, the predicted value is output by Equation (9).

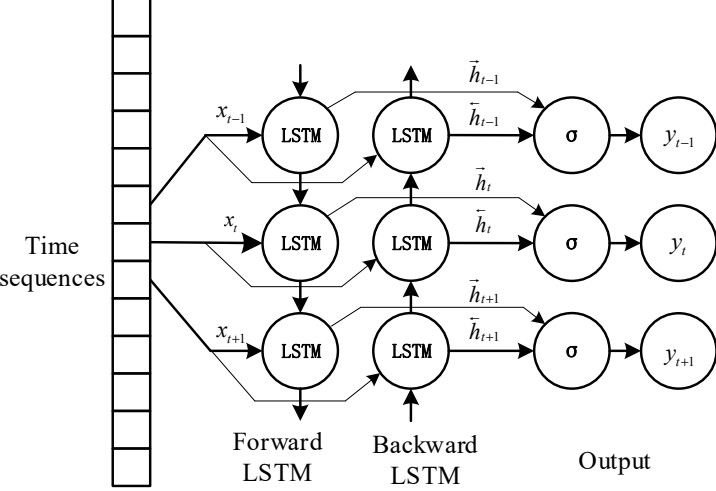

**Figure 3.** Structure of BiLSTM.

### 3.4.3. Muti-Head Self-Attention in the Transformer

The Transformer model [21] is an autoregressive generative model that uses mainly self-attentive mechanisms and sinusoidal location information. Each layer includes a time-self-noticing sublayer, a pre-feedback network sublayer, a residual network sublayer, and a dropout layer.

Attention essentially assigns a weighting factor to each element in the EEG sequence, and if each element is stored, attention can be calculated as the similarity between $Q$ and $K$. The similarity calculated from $Q$ and $K$ reflects the importance of the extracted $V$ values, the weights, which are then weighted and summed to obtain the attention values. The special point of the self-attention mechanism in the $K$, $Q$, $V$ model is that $Q = K = V$. The scaled dot product attention formula is defined as Equation (10):

$$Attention(Q, K, V) = softmax(\frac{QK^T}{\sqrt{d_k}})V \tag{10}$$

The multi-headed self-attention mechanism obtains different representations of h (i.e., each head) of $(Q, K, V)$, calculates the self-attention of each representation, and connects the results. This can be expressed in the same notation as in Equations (11) and (12):

$$head_i = Attention(QW_i^Q, KW_i^K, VW_i^V), \tag{11}$$

$$MultiHead(Q, K, V) = Contact(head_i, \dots, head_h)W^0, \tag{12}$$

where $W_i$ and $W^0$ make the parameter matrix; the structure of the multi-headed self-attentive mechanism is shown in Figure 4.

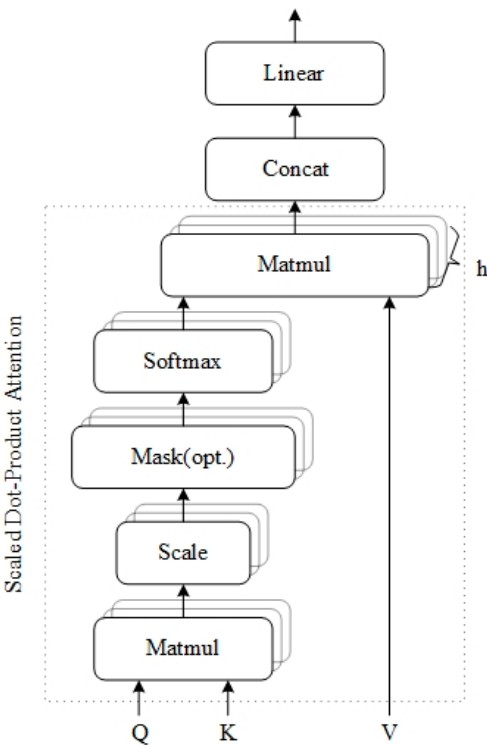

**Figure 4.** Multi-Head Self Attention structure.

The BiLSTM contains two outputs, the output $O = [O_1, O_2, O_3, \dots, O_D]$ for all time steps and the hidden state $H_D$ for the last time step $D$. Since $O = [O_1, O_2, O_3, \dots, O_D]$ denotes the EEG sentiment feature and $H_D$ denotes the hidden temporal feature, to identify the importance of time on sentiment, we need to establish the self-attention relationship between $H_D$ and $O_t$. That is, to establish the weight of each time step output $O_t$ for $H_D$.

Since BiLSTM itself considers the location information, there is no need to set up additional location encoding. The implementation method of the self-attention mechanism in BiLSTM we use is scaled dot product attention [21], which is the self-attention implementation method proposed by the transformer.

The output $O_t$ of each time step is linearly transformed as $Key_t$ and $Value_t$, while the output $H_D$ of the final time step is multiplied by the matrix $W_Q$ as *Query*. at time step $t$, $Key_t$, $Value_t$, $Query$, score $e$ and weight a have the following equations:

$$Query = \omega_Q H_D, \tag{13}$$

$$Value_t = \omega_V O_t, \tag{14}$$

$$Key_t = \omega_K O_t, \tag{15}$$

$$e_t = Query Key_t^T / \sqrt{d_k}, \tag{16}$$

$$a_t = \frac{exp(e_t)}{\sum_{t=0}^{n} exp(e_t)}, \tag{17}$$

where *Query* does not change with time step and $\omega_Q$, $\omega_K$, $\omega_V$ are parameters of the neural network that are modified with backpropagation. The weights $a_t$ and $Value_t$ of each time step are weighted and summed to obtain the emotional feature vector with self-attentiveness:

$$z(Q, K, V) = \sum_t a_t Value_t \tag{18}$$

To obtain the multi-headed self-attention, the above equation is performed h times to obtain the multi-headed self-attention feature $z_1, \ldots, z_h$, which is spliced and linearly transformed once as the final output:

$$MultiHead(Q, K, V) = Concat(z_1, \ldots, z_h)\omega_z \tag{19}$$

*3.5. Training the CNN-BiLSTM-MHSA*

In this paper, the network training is based on cross-entropy function optimization and stochastic gradient descent for backpropagation. The optimizer used for the network is the optimizer responsible for the backpropagation process. The commonly used optimizers are SGD and Adam, both of which are interchangeable. Here, we will use Adam. For backpropagation, we will define a loss function using cross-entropy loss, which is a common loss function used for multi-class purposes. Weight sharing in CNN usually results in varying the gradients across layers, for this reason, single-layer convolutional neural networks are used for binary classification and use a smaller learning rate. Since the two-layer bidirectional LSTM has a larger depth, a smaller number of iterations (epoch = 50) is sufficient to converge, while dropout = 0.5 is adopted in the training network to eliminate the influence of the overfitting problem. The convolutional layer uses a single-layer CNN with a kernel of 128. The convolutional network is used to extract the frequency domain features of the EEG information sequence, and then the LSTM unit analyzes the time domain features. Finally, the multi-head self-attention mechanism is used to improve classification accuracy. In this method, the bidirectional LSTM layer has 256 units (512 in total), and the hidden layer with 128 vs. 512 units is also investigated to select the 256 units with the best results. The multi-head self-attentive mechanism of the head selects the 8 heads with the best effect from 2/4/8/12/16. The following table lists the hyperparameters of the training network. The hyperparameters of the training network are shown in Tables 1–3.

**Table 1.** Hyperparameters of CNN.

| Hyperparameters | Value |
|---|---|
| Kernel | 128 (1D), 3 (2D) |
| Stride | 1 |
| Pool-Kernel | 3 |
| Activation function | Relu |
| Epoch | 50 |
| Padding | 1 |

**Table 2.** Hyperparameters of BiLSTM.

| Hyperparameters | Value |
|---|---|
| Optimizer | Adam (adaptive moment estimation) |
| Input | 32 |
| Hidden | 256 |
| Output | 1 |
| Initial learning rate | 0.0001 |
| Dropout | 0.5 |

**Table 3.** Hyperparameters of Multi-Head Self Attention Structure.

| Hyperparameters | Value |
|---|---|
| Head | 8 |
| K | (Hidden $\times$ 2)/Heads = 64 |

### 3.6. Effect of Different Parameters

In this section, we analyze the impact of the parameter Head on the performance, which is the number of self-attention heads in the attention-based encoder. Table 4 reports the performance of the proposed CNN-BiLSTM-MHSA on the DEAP dataset for different values of Head. This illustrates that the CNN-BiLSTM-MHSA performance does not fluctuate significantly due to the change of the parameter Head, which indicates that the method has relatively good robustness. In addition, when Head is set to 8, the model obtains more competitive recognition results than any other setting. Therefore, to obtain better performance, we set the CNN-BiLSTM-MHSA model to Head = 8. We also compare the hidden layer of the LSTM with the epoch count as in Tables 5 and 6. Different parameter configurations will have an impact, so the most suitable parameters are selected after several experiments.

**Table 4.** Comparison of ablation experiments with different Head.

| Head | Test Accuracy (%) (Binary Classification) | F1-Score | Test Accuracy (%) (Four Classification) |
|---|---|---|---|
| 2 | 93.89 | 92.77 | 87.59 |
| 4 | 93.78 | 92.35 | 87.23 |
| 8 | 94.58 | 93.61 | 88.10 |
| 12 | 93.99 | 92.69 | 87.79 |
| 16 | 94.49 | 93.65 | 87.98 |

**Table 5.** Comparison of different LSTM hidden layer ablation experiments.

| Hidden | Test Accuracy (%) (Binary Classification) | F1-Score | Test Accuracy (%) (Four Classification) |
|---|---|---|---|
| 64 | 90.79 | 89.77 | 83.33 |
| 128 | 92.68 | 91.35 | 84.34 |
| 256 | 94.58 | 93.61 | 85.57 |
| 512 | 93.97 | 91.66 | 85.12 |

**Table 6.** Comparison of different epoch times ablation experiments.

| Head | Test Accuracy (%) (Binary Classification) | F1-Score | Test Accuracy (%) (Four Classification) |
|---|---|---|---|
| 50 | 94.58 | 93.61 | 85.57 |
| 100 | 94.43 | 92.77 | 84.23 |
| 150 | 93.98 | 93.12 | 84.10 |
| 200 | 94.33 | 93.07 | 83.79 |

*3.7. Evaluation Indicators*

Classification accuracy Acc and F-score are used to evaluate the CNN-BiLSTM-MHSA model. Acc is denoted as:

$$Acc = \frac{TP + TN}{TP + TN + FP + FN}, \tag{20}$$

where $TP$ is the number of low-state samples for which the classification model can accurately identify arousal, potency, dominance, and preference; $TN$ is the number of high-state samples for which the classification model can accurately identify arousal, potency, dominance, and preference; $FP$ is the number of low-state misclassifications, and $FN$ is the number of high-state misclassifications.

The precision *Pre* is defined as:

$$Pre = \frac{TP}{TP + FP}, \tag{21}$$

Recall *Rec*:

$$Rec = \frac{TP}{TP + FN}, \tag{22}$$

The *F-score* is an extension of classification accuracy, combining accuracy and recall, and is calculated as defined by:

$$F - score = \frac{2 \times Pre \times Rec}{Rec + Pre} \tag{23}$$

## 4. Materials and Experiments

*4.1. Dataset*

The current experiment is a DEAP dataset consisting of two parts of emotion recognition and physiological signals [31], while the acquisition process is shown in Figure 5. The DEAP dataset collected physical signals and emotional assessments from 32 subjects. Subjects autonomously rated videos from 1 to 9 on 4 dimensions based on arousal, valence, dominance, and liking [32]. The official provides two data forms, one is the original signal including various interferences, such as EMG, EOG, and other interference; the other is preprocessed data, including down-sampling data to 128 Hz, removing EMG artifacts, 4.0–45 Hz band-pass frequency filter filtering, etc. Preprocessing removes the non-emotional information from the previous 2 s to avoid affecting the training of the subsequent models.

The states of valence, arousal, dominance, and liking in the DEAP dataset can be classified according to a given threshold. Emotion states were studied in a quadratic dichotomy of high and low valence, high and low arousal, high and low dominance, and high and low liking with a threshold of 5. The two-dimensional model with four categories consists of a combination of four different affective states: high arousal and low valence (HALV), high arousal and high valence (HAHV), low arousal and low valence (LALV), low arousal and high valence (LAHV) [33].

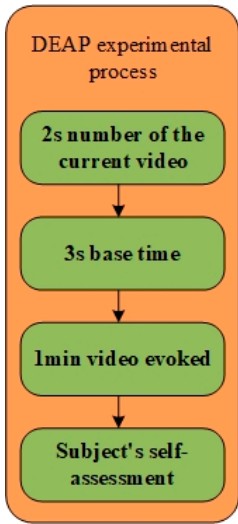

**Figure 5.** DEAP experimental process.

*4.2. EEG Emotion Classfication Experiment*

For Experiment 1, which did not use continuous wavelet transform, the process is shown in Figure 6. We tried to improve the input long-short time memory network before part of the BiLSTM by adding CNN before the BiLSTM, because CNN can help us smooth the signal and will make the sequence length of the EEG signal smaller, so it can greatly the CNN processed the EEG signal after the data as if it was sampled again, instead of using the original data directly, and feed the processed data into BiLSTM to obtain higher accuracy of sentiment classification. In the CNN network, we also add a down sampling, or pooling, layer, which reduces the parameters and speeds up the computation. The data length will not change. Then we can use BiLSTM to learn the temporal features passed by CNN in a more in depth way. LSTM can only consider the sentiment information in a shorter period of time, while BiLSTM can capture the influence of the future and past on the present moment, learn the future and past information of the time series, and be able to concatenate the sentiment features extracted before and after as the final sentiment features. However, BiLSTM may not take advantage of the association of given labels, such as whether high valence has an effect on low valence, or the association between other criteria given in more classifications. The multi-headed attention mechanism incorporated in the experiments of this paper solves this problem well. The features of all time steps and hidden states of the BiLSTM can be weighted and reassigned using the multi-headed self-attentive mechanism by back-propagating the parameters of Value and Key over time to modify the parameters and obtain the emotional feature vector of attention based on the weighted sum of the calculated time steps and Value to obtain the multi-headed state. Attention reassignment achieves weighting the key information of EEG emotion features and improves the emotion recognition accuracy.

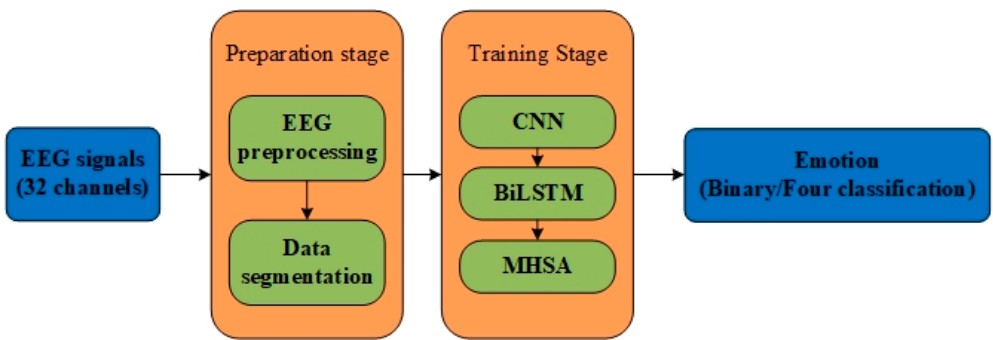

**Figure 6.** CNN-BiLSTM-MHSA training process.

The proposed CNN-BiLSTM-MHSA model is used for the identification of high and low potency, arousal, control, and preference in the DEAP dataset. The dataset python pre-processed version was selected for this experiment. The first 32 EEG channels were extracted and the data were downsampled to 128 Hz. Thus, one second includes 128 samples, and one minute or 60 s about 7680 samples, but the 8064 samples in the data should be $128 \times 63$ samples, with 3 s of benchmark time for each test. Data segmentation is used to create more data. In this paper, a 1-min video is divided into 12 five-second segments, which greatly increases the number of samples and thus increases the chance of prediction. The original data format is $1280 \times 32 \times 8064$. After dividing into 12 segments, the data format is $1280 \times 32 \times 672 \times 12$, and then the data is replaced with $15,360 \times 32 \times 672$. The data are fed into a CNN-BiLSTM-MHSA network, where the batch size is 16. The leave-one-subject-out cross-validation method is used, and the high and low states are distinguished by a threshold value of 5 afterwards, with the high label set to 1 for those greater than 5 and the low label 0 for the rest. Training and validation accuracies of 99.07% and 95.12% were obtained on the DEAP dataset for the potency state, respectively. The training and validation accuracies in the arousal state were 99.12% and 94.62%. The training and validation accuracies were 99.25% and 94.32% for the dominance state, and 99.02% and 94.25% for the preference state, respectively.

In the validating split process, this paper defines the training and validation ratio as 7:2, and the remaining data is used as a test model. The training steps are repeated for each emotional state (valence, arousal, dominance, and liking), and the classification training result graph is shown in Figure 7. All training and testing procedures proposed in this study were performed on a server with 4 Nvidia GeForce RTX2080Ti GPUs and 256 GB RAM by using Cuda 11.2, Pytorch version 1.11.

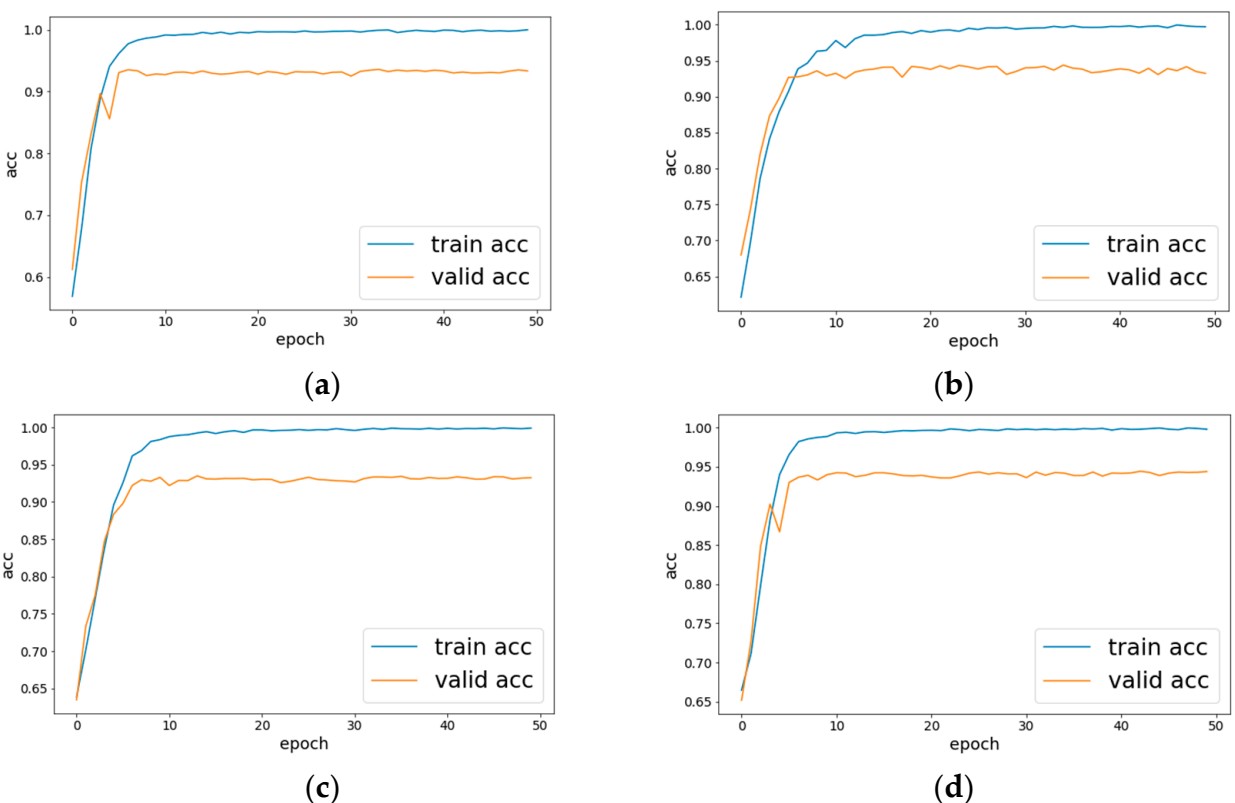

**Figure 7.** (**a**) Valence classification training results, (**b**) arousal classification training results, (**c**) dominance classification training results, (**d**) liking classification training results improve the training speed.

Experiment 2 using the time–frequency diagrams of CSP and CWT as input data is shown in Figure 8, and the method used is referred to as CP-CNN-BiLSTM-MHSA. The images are used as data input to CNN-BiLSTM-MHSA, and the input to CNN is a $20 \times 65 \times 3$ time–frequency map. To perform better feature extraction of the image, the CNN in this paper is designed as a two-dimensional convolution of two layers, the original data $1280 \times 32 \times 8064$ is transformed into $7680 \times 32 \times 20 \times 65$ and then sent into the CNN in the form of padding after the convolution operation to obtain the dimension $7680 \times 64 \times 20 \times 65$, and then after pooling down sampling it becomes $7680 \times 64 \times 10 \times 32$ and when sent into the next CNN layer and pooling layer becomes $7680 \times 128 \times 5 \times 16$. The whole training process has 102,977 training parameters. The training is based on cross-entropy function optimization and back-propagation of stochastic gradient descent for network training, and the optimizer continues to use Adam with a learning rate of 0.0001 and 50 iterations. The average accuracy of training and testing in high and low valence, arousal, dominance, and liking states was 99.98 and 98.10% respectively. Their other performance index F1 score values were 98.22%, 98.02%, 98.17%, and 98.11% respectively. The training accuracy is improved by about 4% compared to the experiments with the time series of input EEG, and the training accuracy and test accuracy shown in Figure 9.

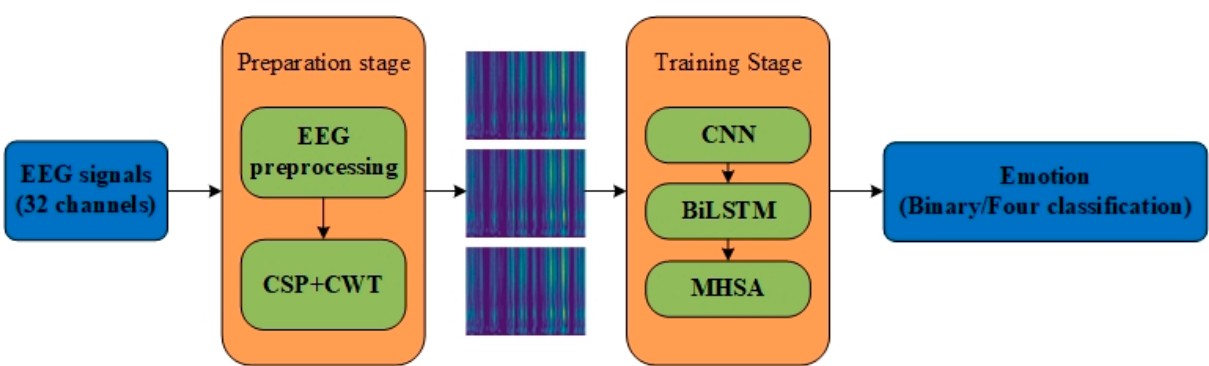

**Figure 8.** Multi-Head Self Attention structure.

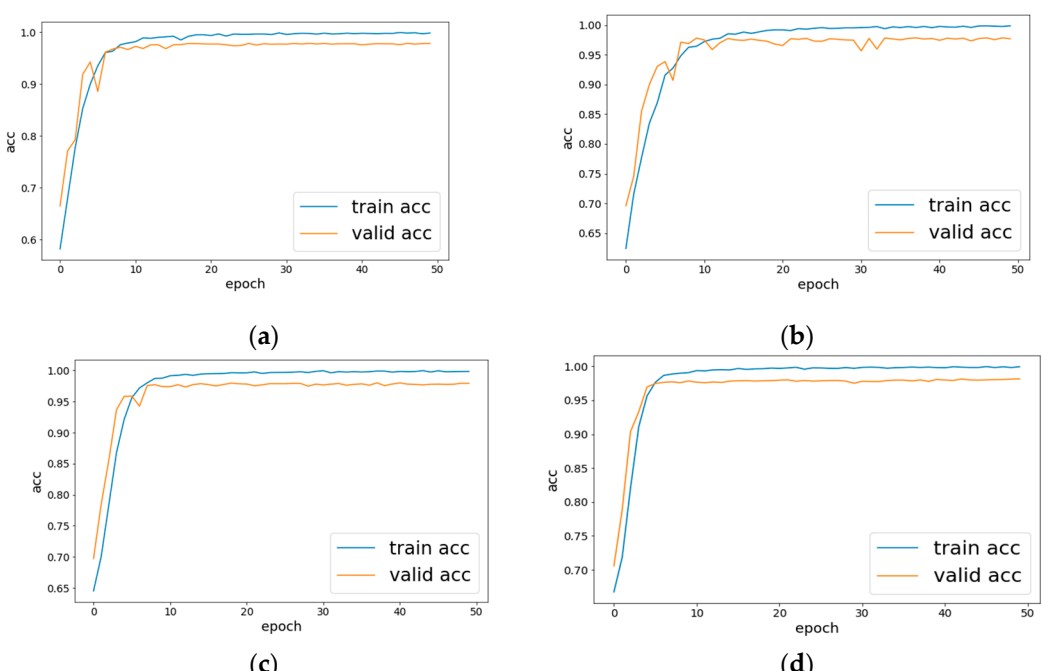

**Figure 9.** (**a**) Valence classification training results, (**b**) arousal classification training results, (**c**) dominance classification training results, (**d**) liking classification training results.

### 4.3. EEG Emotion Four-Classification Experiment

In this paper, four classes of high and low valence-arousal (HALV, HAHV, LALV, LAHV) were studied. The first 32 channels of data were selected, and the total duration of the data experiment was 40 min, all of which yielded 1280 labels, classified using HALV, HAHV, LALV, and LAHV labels. Based on the positive and negative deviations of potency and arousal, we mapped each trial into four quadrants to form an effective classification label. It was possible to classify HALV = 352, HAHV = 348, LALV = 282, and LAHV = 298. The segmented data and labels are fed into the CNN-BiLSTM-MHSA network with a training and validation ratio of 7:2, using the cross-validation method. The scored data and labels were fed into the CNN-BiLSTM-MHSA network with a training to validation ratio of 7:2, using a cross-validation method, and the average validation and testing accuracy of the four classifications for high and low validity and arousal was 89.58 and 85.57%; the training results are shown in Figure 10a.

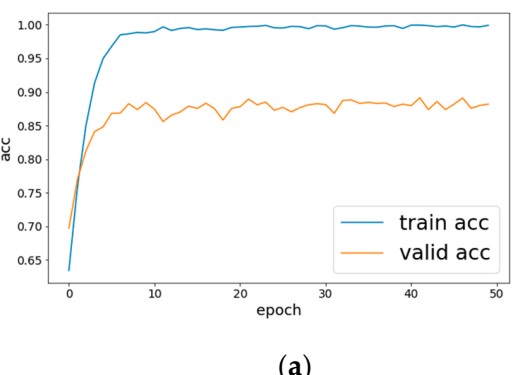
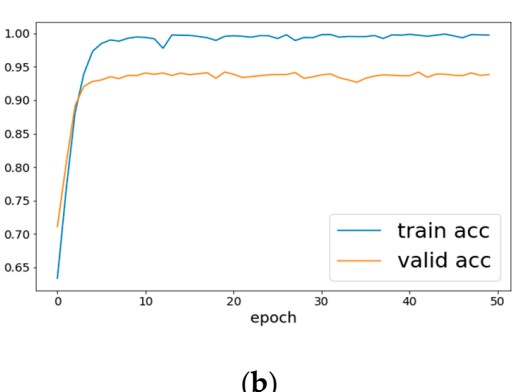

|                          |                          |
| :----------------------: | :----------------------: |
|           (**a**)        |          (**b**)         |

**Figure 10.** (**a**) represents the training and validation accuracy for the four classification tasks in Experiment 1, with the input data as EEG sequence information. (**b**) represents the training and validation accuracy for the four classification task in Experiment 2, with the input data as a $20 \times 65 \times 3$ time-frequency plot.

The second approach in high and low valence-arousal four classification investigates the use of CWT to extract features from the time–frequency maps of CSP spatially filtered signals for classification, with an input of $20 \times 65 \times 3$ time–frequency maps. The network is trained based on cross-entropy function optimization and back-propagation stochastic gradient descent, with an optimizer using Adam, a learning rate of 0.0001, and several 50- iterations. The average accuracy of validation and testing in high and low valence-arousal states was 92.89 and 89.33% respectively. The training accuracy is improved by 4% compared to the training accuracy of the time series of the input EEG. The results are shown in Figure 10b.

### 4.4. Experimental Comparison

In this paper, the proposed CNN-BiLSTM-MASH fusion model is compared in ablation experiments on DEAP dataset by fusing convolutional neural network, long and short-term memory network, and attention mechanism with each other. The compared models are LSTM, BiLSTM, CNN-LSTM, CNN-BiLSTM, CNN-BiLSTM-Attention, and CNN-BiLSTM-MASH. all models are used on the DEAP dataset to compare the models on datasets for testing accuracy and time complexity of training and testing. The datasets in all experiments were divided into 7:2:1 patterns and tested on the server. The architecture has a training time of 88.57 s for one epoch on the dataset and a testing time of 11.41 s for the same cycle, and all the results are shown in Table 7.

**Table 7.** Comparison with the base model.

| Methods | Recognition | Accuracy (%) | Training Time (s) | Testing Time (s) |
|---|---|---|---|---|
| LSTM | H/L Valence<br>H/L Arousal<br>H/L Dominance<br>Like/Unlike State | 71.53 | 105.20 | 13.23 |
| BiLSTM | H/L Valence<br>H/L Arousal<br>H/L Dominance<br>Like/Unlike State | 75.45 | 109.18 | 12.96 |
| CNN-LSTM | H/L Valence<br>H/L Arousal<br>H/L Dominance<br>Like/Unlike State | 84.13 | 90.45 | 11.59 |
| CNN-BiLSTM | H/L Valence<br>H/L Arousal<br>H/L Dominance<br>Like/Unlike State | 90.12 | 95.05 | 12.06 |
| CNN-BiLSTM-Attention | H/L Valence<br>H/L Arousal<br>H/L Dominance<br>Like/Unlike State | 92.25 | 86.20 | 11.47 |
| CNN-BiLSTM-MHSA | H/L Valence<br>H/L Arousal<br>H/L Dominance<br>Like/Unlike State | 94.58 | 88.57 | 11.41 |

It can be seen that only one layer of the LSTM network works poorly because a single sample point is only a single data point at the time of input. Since most of the EEG signals are long, the DEAP dataset used in this paper, for example, has a signal length of 40 min and 8064 sampling points. Therefore, the single-layer LSTM cannot understand the relationship between the 1st second (sample point 1) and 60 s (sample point 8064) after the input sequence. The subsequent addition of the CNN network combined with LSTM brings a huge improvement, which means that the smoothing and convolutional signals of the 1D convolutional network have better performance for the training of LSTM, thus solving the problem of long-time sequence training of LSTM. While the addition of a bidirectional two-layer LSTM structure allows the before-and-after time series to be understood, improving the accuracy by about 6%, the addition of an attention mechanism on top of it brings a slight improvement, increasing the accuracy by about 2%. In this paper, we propose a fusion model combining one-dimensional convolution, a bi-directional two-layer LSTM, and a multi-headed self-attentive mechanism. Among them, BiLSTM can utilize both earlier and later sequence information, which helps to explore deep cognition from EEG sequence signals, while the addition of a multi-headed self-attention mechanism is superior to single-headed attention. The CNN-BiLSTM-MHSA model studied in this paper achieves an average accuracy of 94.58% for quadratic binary classification, and the comparison graph is shown in Figure 11.

The results of the dichotomous classification experiments were compared with other dichotomous classification studies using DEAP databases or CNN and DNN types, and the corresponding results are given in Table 8. Compared with single CNN and DNN [16,34], the method in this paper shows a great improvement, which demonstrates that dynamic temporal features have a great impact on the accuracy of sentiment recognition, and single spatial or temporal features are poorly recognized. The key information of past and future in temporal dynamic features is also important. The CNN-LSTM [17] has no learning of

future emotional states in EEG time series signals, accuracy is still low. The DE-CNN-BiLSTM [12] fully considers the complexity and spatial structure of the brain and considers the temporal properties of dynamic EEG signals. However, there is no learning of time–frequency features, and no self-attention mechanism is used to redistribute the weights of EEG emotional key information. There is no convolutional smoothing signal processed by CNN in LSTM-Attention [14], no resolution of spatial structure, just resolution of a single time series with attention. In contrast to these studies, the CP-CNN-BiLSTM-MHSA method proposed in this paper improves the spatial information in the EEG signal, and the features in the time–frequency domain are analyzed and then input to the network as a time–frequency map. This feature information was re-extracted using CNN, for the dynamic temporal features present in the EEG, and then the future and past key sentiment information in it was fully learned using BiLSTM. Finally, the weight of this information is redistributed using a self-attentive mechanism, making the network more capable of recognizing this information. In the comparison of binary classification in the DEAP dataset, the proposed method in this paper is significant in sentiment recognition research enhancement.

The results of the four-classification experiment were compared with the previous two-dimensional potency-arousal four-classification studies of HALV, HAHV, LALV, and LAHV using the DEAP dataset, and the results are shown in Table 9. The CNN-LSTM [18] fuses the spatial, frequency domain, and temporal features of the original EEG signal, but the recognition accuracy is still low for the four-classification tasks. The SVM [22–24] used wavelets to decompose and extract the time–frequency features and smoothed the features into the SVM but did not improve the spatial information and dynamic time features of the EEG. The PSO-BiLSTM [19] deep learning techniques based on long and short-term memory are used to retrieve emotional changes from optimized data corresponding to labeled EEG signals and to remove repetitive information, but this also lacks learning of spatial and time–frequency features. The WP-KNN [13], on the other hand, does not analyze the dynamic time features. In summary, the method proposed in this paper fully analyzes the dynamic temporal features and spatial information present in EEG and is effective in four-category sentiment recognition.

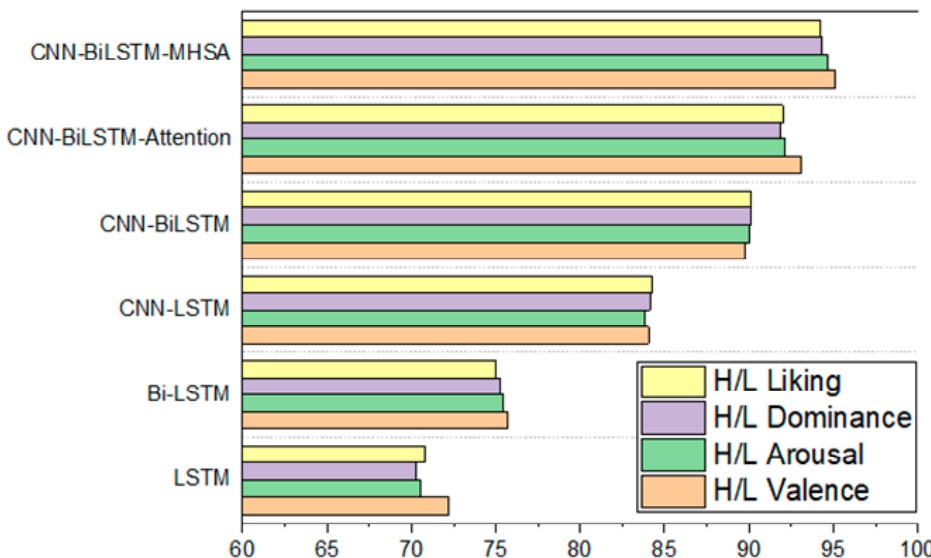

**Figure 11.** Model comparison chart.

**Table 8.** Comparison with previous two-classification studies.

| Study | Recognition | Dataset | Training Accuracy (%) | Test Accuracy (%) | F1-Score (%) | Methods |
|---|---|---|---|---|---|---|
| Tripathi [16] | H/L Valence<br>H/L Arousal | DEAP | N/A | 73.16<br>81.78 | N/A | CNN |
| Nafjan [34] | H/L Valence<br>H/L Arousal | DEAP | N/A | 82.00 (avg) | N/A | DNN |
| Jinpeng [17] | Negative/Neutral/Positive Beta Gamma | SEED | 98.00 (avg) | 86.20<br>88.20 | N/A | HCNN, SAE |
| Ozdemir [9] | H/L Valence<br>H/L Arousal<br>H/L Dominance<br>Like/Unlike State | DEAP | 94.07<br>92.02<br>92.85<br>92.02 | 90.62<br>86.13<br>88.48<br>86.23 | 89.98<br>88.01<br>90.76<br>90.27 | CNN-LSTM |
| Cui F [12] | H/L Valence<br>H/L Arousal<br>Negative/Neutral/Positive Beta Gamma | DEAP<br>—<br>SEED | N/A | 94.02<br>94.86<br>—<br>94.82 (avg) | N/A | DE-CNN-BiLSTM |
| Kim Y [14] | H/L Valence<br>H/L Arousal<br>H/M/L Valence<br>H/M/L Arousal | DEAP | N/A | 90.10<br>87.90<br>86.90<br>84.4 | N/A | LSTM-Attention |
| The proposed study 1 | H/L Valence<br>H/L Arousal<br>H/L Dominance<br>Like/Unlike State | DEAP | 99.07<br>99.12<br>99.25<br>99.02 | 95.12<br>94.62<br>94.32<br>94.25 | 94.77<br>94.46<br>94.53<br>94.66 | CNN-BiLSTM-MHSA |
| The proposed study 2 | H/L Valence<br>H/L Arousal<br>H/L Dominance<br>Like/Unlike State | DEAP | 99.98 (avg) | 98.10 (avg) | 98.22<br>98.02<br>98.17<br>98.11 | CP-CNN-BiLSTM-MHSA |

**Table 9.** Comparison with previous four-classification studies.

| Study | Recognition | Dataset | Test Accu-Racy (%) | Methods |
|---|---|---|---|---|
| Li et al. [18] | H/L Valence/Arousal | DEAP | 75.12 | CNN-LSTM |
| Gupta et al. [22] | H/L Valence/Arousal<br>Negative/Neutral/Positive on Beta Gamma | DEAP | 71.43<br>—<br>83.33 (avg) | FAWT-SVM |
| Aguiñaga et al. [23] | H/L Valence/Arousal (3)<br>H/L Valence/Arousal (4) | DEAP<br>—<br>SEED | 84.20 (avg)<br>—<br>80.90 | WP-NN-SVM |
| Ozel et al. [24] | H/L Valence/Arousal | SEED | 76.30 | MSST-SVM |
| Xu et al. [25] | H/L Valence/Arousal | DEAP | 80.83 | mRMR-KELM |
| Sharma et al. [19] | H/L Valence<br>H/L Arousal<br>H/L Valence/Arousal | DEAP | 84.16<br>85.21<br>82.01 | PSO-BiLSTM |
| Galvão [13] | H/L Valence<br>H/L Arousal<br>H/L Valence/Arousal | DEAP | 89.83<br>89.84<br>84.40 | WP-KNN |
| The proposed study 1 | H/L Valence/Arousal | DEAP | 85.57 | CNN-BiLSTM-MHSA |
| The proposed study 2 | H/L Valence/Arousal | DEAP | 89.33 | CP-CNN-BiLSTM-MHSA |

## 5. Results

In this paper, we classify EEG emotions in four dimensions, namely potency, arousal, dominance, and preference. A deep learning approach is proposed to analyze emotional states using deep learning. We proposed a CNN-BiLSTM-MASH model for EEG emotion classification and also extracted emotion features by scale maps. Experiments were conducted on the DEAP dataset, and it was concluded that the proposed framework achieves better results in experiments with dichotomous and quadruple classifications. The analysis of one-dimensional EEG signals is performed using data segmentation to increase the training number, and after CNN smoothing of the signal as well as pooled down sampling, BiLSTM is used to obtain the future and past correlation features of the sentiment time series. The weighted redistribution of hidden features using multi-headed self-attentiveness improves the EEG sentiment recognition accuracy, and the proposed CP-CNN-BiLSTM-MASH model is used to extract the sentiment features by CSP filtered scale maps to extract emotion features, which has a good effect on the four classification results.

In recent years, with the development of deep learning technology, the models applied to EEG emotion recognition have become more and more diverse. In this paper, we propose a new fusion deep learning model with new enhancements on the DEAP dataset, but there is still much room for improvement in four-category recognition. In the future, we plan to conduct improved deep training in this area.

**Author Contributions:** Data curation, L.C.; Investigation, Z.H. and Y.L.; Resources, Z.H. and Y.L.; Software, L.C. and J.Z.; Supervision, Y.L.; Writing—original draft, L.C.; Writing—review & editing, L.C. All authors have read and agreed to the published version of the manuscript.

**Funding:** This research was funded by the National Natural Science Foundation of China (No.51775076).

**Institutional Review Board Statement:** Not applicable.

**Informed Consent Statement:** Not applicable.

**Data Availability Statement:** Not applicable.

**Acknowledgments:** Publicly available datasets were analyzed in this study. This data can be found here: [http://www.eecs.qmul.ac.uk/mmv/datasets/deap/] (accessed on 1 October 2022).

**Conflicts of Interest:** The authors declare no conflict of interest.

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
