# Peer review of "EEG-Based Emotion Recognition Using Convolutional Recurrent Neural Network with Multi-Head Self-Attention"

_applsci, doi:10.3390/app122111255_

Round 1

Reviewer 1 Report

In this paper convolutional recurrent neural network with Multi-Head Self-Attention is proposed using Electroencephalogram signals data for motion recognition. The paper is interesting and written well. However, there are a few major issues that need to be resolved before this manuscript can be accepted.

1)      The abstract has to be changed because it doesn't adequately explain the study topic and research question.

2)      Motivation is not clear.

3)      Research gap is not defined.

4)      I suggest the authors to write their main contributions in bullets form in the introduction section.

5)      Paper structure paragraph is missing in the introduction section.

6)      Time series analysis is a challenging task in different domain. However, a discussion on it is missing in the introduction and literature section. Following is a recent paper related to time series analysis: (https://doi.org/10.3390/math9243326)

7)      Implementation and system details are missing, which is very important for the reader to produce similar results.

8)      It is crucial to explain how the machine learning algorithms' hyper-parameters are configured. How can we be certain that the techniques' accuracy won't be impacted by parameter tuning?

9)      I suggest the authors to provide the comparative complexity analysis of the proposed method with the other methods.

10)  The language is poor and needs polishing.

Author Response

Dear reviewer:

Thank you for your decision and constructive comments on my manuscript. We have carefully considered the suggestion of Reviewer and make some changes. We have tried our best to improve and made some changes in the manuscript.

The revised item-by-item description has been modified according to your comments as follows:

(1) I have rewritten the abstract to restate the research topic and problem;

(2) I also rewrote the introduction to highlight my motivation;

(3) I also reworked the related work;

(4)-(5) I have added this to the introduction;

(6) EEG signal sampling is time series information, so I did not highlight it before, LSTM does have excellent ability to handle time series, many papers I cited also used this, thank you for recommending the relevant papers, I also cited this article to highlight the excellent processing of time series by LSTM;

(7) I redesigned the experimental part of the training introduction;

(8) I added different effects of hyperparameters;

(9) I added time complexity comparison, which is very important in human-computer interaction;

(10) I rewrote most of the article.

Reviewer 2 Report

The topic addressed in this paper is definitely interesting, and the proposed hybrid solution combining a convolutional neural network (CNN), a bidirectional long short-term memory network (BiLSTM), and a multi-head self-attention mechanism (MHSA) to improve the accuracy of the emotion recognition starting from the EEG signals might indeed be valuable.

However, the presentation abounds in sentences that sound very general and don't have much meaning (e.g. lines 58-64, 113-120, 171-174 etc.).

In this context, I would cite the phrase found in line 534, which sounds: "A deep learning approach is proposed to analyze emotional states using deep learning".

In my opinion, the presentation of the context of this work seems superficial. Maybe consulting the following comprehensive reviews of the state of the art in this field, might be helpful:

Liu, H., Zhang, Y., Li, Y., & Kong, X. (2021). Review on Emotion Recognition Based on Electroencephalography. Frontiers in Computational Neuroscience, 84.

Craik, A., He, Y., & Contreras-Vidal, J. L. (2019). Deep learning for electroencephalogram (EEG) classification tasks: a review. Journal of neural engineering, 16(3), 031001.

 Maithri, M., Raghavendra, U., Gudigar, A., Samanth, J., Barua, P. D., Murugappan, M., ... & Acharya, U. R. (2022). Automated emotion recognition: Current trends and future perspectives. Computer Methods and Programs in Biomedicine, 106646.

Same remark for the section Method.

The text formatting also needs attention - for example, the text in lines 380-418 is a single paragraph, and obviously is difficult to read and understand.

Also, note that the paragraphs 3.4.1 and 3.4.2 have the same title.

Author Response

Dear reviewer:

Thank you for your decision and constructive comments on my manuscript. We have carefully considered the suggestion of Reviewer and make some changes. We have tried our best to improve and made some changes in the manuscript.

The revised item-by-item description has been modified according to your comments as follows:

(1) Abstract, introduction, and related work I have reworked;

(2) I have also reworked the background of the work;

(3) The same tables were replaced as well;

(4) the formatting in the experiments has also been adjusted.

(5) the headings of 3.4.1 and 3.4.2 as well as replacements.

Round 2

Reviewer 1 Report

I think the paper is suitable for publication in its present form. 

Reviewer 2 Report

It seems that the authors have massively re-worked the presentation, which is now, in principle acceptable for publication. There are still some issues with the English language, but these can be solved in the editing phase.